# Awareness and Implementation of Sports Injury Prevention Programs Among Physical Therapists in Saudi Arabia: A Cross-Sectional Study

**DOI:** 10.3390/medicina61010121

**Published:** 2025-01-14

**Authors:** Yousef M. Alnefaie, Mohamed K. Seyam, Msaad Alzhrani, Ahmad Alanazi, Faris S. Alzahrani, Saud M. Alsaadoon, Shahnaz Hasan

**Affiliations:** 1Department of Rehabilitation, King Abdulaziz Medical City, Riyadh 11426, Saudi Arabia; alnufaieyo@mngha.med.sa (Y.M.A.); alsaadoonsa@mngha.med.sa (S.M.A.); 2Department of Physical Therapy and Health Rehabilitation, College of Applied Medical Sciences, Majmaah University, Al Majmaah 11952, Saudi Arabia; m.seyam@mu.edu.sa (M.K.S.); m.alzhrani@mu.edu.sa (M.A.); aalanazi@mu.edu.sa (A.A.); 3Department of Physical Therapy and Health Rehabilitation, Ministry of Health, King Saud Hospital, Onaizah 56242, Saudi Arabia; falzahrani103@moh.gov.sa

**Keywords:** awareness, implementation, sports injury prevention, Saudi Arabia

## Abstract

*Background and Objectives*: Sports injury prevention programs (SIPPs) are crucial for mitigating sports injuries and enhancing athletes’ performance. In Saudi Arabia, the sports sector is growing, and the awareness and implementation of sports injury prevention programs (SIPPs) among physical therapists require examination. This study aims to evaluate physiotherapists’ awareness of and the implementation of sports injury prevention programs (SIPPs) in the Saudi Arabian region with findings that could enhance rehabilitation and sports injury prevention practices. *Materials and Methods*: A self-administered online questionnaire was distributed to licensed physiotherapists in Saudi Arabia. Three hundred sixty-six participants responded to the questionnaire, of whom 55.5% were male and 44.5% were female physiotherapists. *Results*: Licensed physical therapists show a high awareness of sports injury prevention, with 83.9% agreeing or strongly agreeing. However, only 53.8% were aware of sports injury prevention programs, and 37.7% reported actively implementing them. Sports physical therapists scored significantly higher in awareness and implementation than other specialties (*p* < 0.001) with no significant regional differences. Educational qualification was also significant, with those holding a master’s degree or higher reporting greater awareness and implementation than those with a bachelor’s degree (*p* = 0.007). There was a strong positive correlation between awareness and implementation (r = 0.723, *p* < 0.01), along with weak correlations between awareness and perceived barriers (r = 0.270, *p* < 0.01) and implementation and barriers (r = 0.280, *p* < 0.01). *Conclusions*: This study finds that physical therapists in Saudi Arabia have moderate-to-low awareness and implementation of sports injury prevention programs (SIPPs), especially outside of sports-specific fields. Sports physical therapists and those with higher education have significant awareness. There are minimal regional differences but a strong positive correlation between awareness and implementation. Enhanced training, resources, and institutional support are needed to improve SIPP implementation in rehabilitation.

## 1. Introduction

Sport and recreation are commonly advocated as integrated components that promote physical, mental, and emotional well-being across all demographics [1,2]

In Saudi Arabia, there is a growing interest in the sports sector and ongoing investments in sports infrastructure and professional leagues. This has led to an increasing need for effective sports injury prevention strategies to support recreational and professional athletes in maintaining their physical health and optimizing performance [3,4]. Individuals who engage in these endeavors demonstrate robust emotional and intellectual capacities due to their professional or educational pursuits [5,6]. Insufficient physical activity has been associated with adverse effects on health, such as the development of type 2 diabetes, cardiovascular disease (CVD), stroke, depression, and some forms of cancer. Sports and physical activity significantly benefit individuals and society, reducing lifestyle diseases and decreasing the public health burden [7,8].

A rise in the intensity of physical exercise is linked to a higher vulnerability to sports-related injuries in athletes [9]. Engebretsen et al. define sports injuries as “any damage of the body tissues that occurs as a result of sport or exercise” [10,11]. Sports-related injuries have negatively affected individuals’ engagement in physical activity, including increased risk of post-traumatic osteoarthritis and higher rates of obesity [12].

Research conducted in Arab countries between 1985 and 2015 highlights the prevalence of sports-related injuries, indicating rates as high as 70%, underscoring the need for preventative measures [13,14]. Globally, the economic burden of such injuries is significant, with a study conducted in Australia between 2004 and 2010 estimating the direct financial burden of sports-related injuries. The study reveals the economic burden of sports injuries, showing a total cost of 265 million AUD over seven years, especially related to lower limb injuries [15]. Sports injury prevention programs (SIPPs) reduce the probability of injury incidence, foster athletes’ well-being and physical condition, optimize performance capabilities, and mitigate the expenses linked to healthcare [16].

Developing a sports injury prevention program (SIPP), known as FIFA 11+, was a collaborative effort by the Federation International Football Association (FIFA) Medical Assessment and Research O’Brien (F-MARC), the Oslo Sports Trauma Research Center (Norway), and the Santa Monica Orthopedic and Sports Medicine Center (USA). Implementing an athlete’s three-stage program, consisting of a warm-up activity routine and 15 organized exercises arranged in a specific sequence, is straightforward [17,18,19].

A comprehensive meta-analysis of multiple meta-analyses examining the efficacy of FIFA 11+ injury prevention programs in soccer demonstrated a significant % of overall risk reduction of 34% (RR = 0.66 [0.60–0.73]) for all types of injuries. Furthermore, a notable meta-analysis of meta-analyses observed a reduction of 29% (RR = 0.71 [0.63–0.81]), specifically for lower limb injuries [19].

Before developing the FIFA 11+ program, certain researchers and athletes implemented the Prevent Injury and Enhanced Performance (PEP) program and the Knee Injury Prevention Program (KIPP) [20]. The regimen encompasses a variety of physical activities, such as warm-up exercises, stretching routines, workouts aimed at enhancing strength, plyometric exercises, and drills to improve agility skills [20]. Likewise, it has been demonstrated to be efficacious in preventing or mitigating injury within a broader demographic [21]. These programs include neuromuscular training (NMT), progressive strengthening, balance training, plyometrics, and agility exercises to promote an understanding of the significance of preventing dynamic knee valgus and elevating the risk of anterior cruciate ligament (ACL) injuries [22,23]. Implementing the KIPP program during preseason or in-season warm-ups has been found to result in a reduction in lower extremity injuries among female adolescent soccer and basketball athletes [23]. Despite their proven efficacy, awareness, and implementation of current sports injury prevention programs, improvement is still needed among physiotherapists, health professionals, coaches, and key stakeholders [24,25]. Physiotherapists have the expertise to diagnose, treat, and prevent sports-related injuries and are pivotal in enhancing athletes’ performance, reducing injury risks, and promoting long-term health [24]. A study aimed to assess the physiotherapists’ awareness and implementation of SIPPs from an international perspective. This study’s findings indicate that physiotherapists have average awareness and low-level implementation of SIPPs [26]. Another study aimed to assess the existing perception, knowledge, and practice of sports injury prevention among sports professionals in Western Europe [27]. The findings of this study indicate a need for more awareness regarding injury prevention concepts among physiotherapist professionals within the European regions. The extent of this gap was found to be dependent on the specific professional occupation and the country in which individuals were employed [28].

Therefore, there is a significant research motivation to assess the awareness and implementation of SIPPs among physiotherapists in Saudi Arabia. This research will enhance the literature by providing specialized insights into the awareness and implementation of SIPPs among Saudi physical therapists, improving athletic care, and furthering knowledge in this area, which could improve athletes’ safety and performance and promote a healthy and active lifestyle.

## 2. Materials and Methods

### 2.1. Research Design

This study employed a cross-sectional survey with a convenience sampling method.

The sample size of the survey was calculated using the equation [29] *N* = (1 − *p*) × *z*2*e*21 + (1 − *p*) × *z*2*ne*2
where *N* = population size, *e* = margin of error, *p* = estimated proportion, and *z* = critical value for confidence level.

The population size practicing in Saudi Arabia is 6028 licensed physical therapists obtained from the Saudi Commission for Health Specialties. The sample size was determined with a confidence level of 95% and a margin of error of 0.05, and the targeted sample was found to have a minimum of 362 participants. The licensed physical therapists with all specialties working in Saudi Arabia were eligible to participate in this study. The physical therapy students and interns were excluded. The reason for including all specialties other than sports physical therapists was to gain and assess their insight into the extent of awareness, implementation, and barriers related to SIPP while providing rehabilitation and training. The physical therapy students and interns were excluded. Written permission from the Institutional Review Board (IRB), The Majmaah University for Research Ethics Committee (MUREC)—HA-01-R-088 (Ethics Number: MUREC-June. 19/COM-2023/23-3)—was obtained. Before completing the questionnaires, each participant signed an informed consent form and a cover letter, ensuring complete confidentiality throughout this study.

### 2.2. Developing the Survey and Validity

The survey instrument was designed using a two-fold approach that included an extensive examination of the existing literature and essential input from experts in the field. The integration of both methods ensured that the survey questions were appropriately congruent with the research objectives and also culturally sensitive and contextually relevant for the Saudi culture, making it a suitable fit for the Saudi population. This study examines the awareness and implementation of SIPP among physical therapists in Saudi Arabia. This includes gathering demographic information from participants, such as gender, age, years of professional experience, work settings, highest educational credentials, specialty, and region within Saudi Arabia. The survey focuses on assessing awareness through seven questions, evaluating the implementation of SIPP through seven questions, and identifying potential barriers through four questions—a closed-ended question with a Likert scale. The sole feasible result of responding is limited to the options of strongly agree, agree, neutral, disagree, or strongly disagree.

### 2.3. Content Validity

To achieve content validation, a panel of six experts in sports physical therapy was contacted to participate in an online survey and rate every aspect of the survey according to its relevance to the domain. The minimum selection rate was 83% to be included in the survey [28]. Feedback led to minor adjustments in wording to enhance clarity and cultural sensitivity for the Saudi context, as shown in Table 1.

Based on the above calculation, we conclude that the S-CVI average is based on I-CVI, the S-CVI average is based on proportion relevance, and the S-CVI UA average meets a satisfactory level. Thus, the scale of the questionnaire has achieved satisfactory content validity [28].

### 2.4. Data Collection

Data were collected using an online survey distributed via Google Forms, designed to meet the study’s objectives of assessing awareness, implementation, and barriers to SIPP adoption among licensed physical therapists in Saudi Arabia. The survey link was distributed through email lists, professional associations, and social media platforms related to physical therapy. Participants were given two weeks to complete the survey, with a reminder sent at the midpoint to encourage responses.

The survey instrument was divided into multiple domains to collect relevant data systematically. The primary phase of this study focused on gathering demographic information from the participants. This included gender, age, experience, job environment, highest educational degree, specialization in physical therapy, and region in Saudi Arabia. The subsequent sections included inquiries regarding the participants’ understanding of sports injury prevention programs, their awareness of and the implementation of the SIPPs, and factors that may hinder the implementation.

### 2.5. Data Analysis

The data were analyzed using descriptive and inferential statistical techniques. Descriptive statistics, including means and standard deviations, were computed to summarize the participants’ demographic characteristics, awareness, implementation practices, and potential barriers. The objective of this study is to utilize inferential statistical techniques for analysis of variance (ANOVA) to compare the average scores of awareness, implementation, and potential barriers across different physical therapy specializations and regions in Saudi Arabia. Independent sample *t*-tests assessed differences in awareness and implementation levels based on educational qualifications. A Pearson correlation analysis examined the relationships between key study variables, explicitly exploring the association between awareness, implementation levels, and perceived barriers. The statistical significance was determined at a significance level of *p* < 0.05 (two-tailed). The results are shown in tables and graphs for clarity.

## 3. Results

### 3.1. Reliability of the Scale

To ensure the reliability and suitability of the data for further analysis, the psychometric properties of the major study variables, including awareness, implementation, barriers, and the total scale, were assessed. A reliability analysis yielded strong Cronbach’s alpha values for awareness (α = 0.83), implementation (α = 0.90), and the total scale (α = 0.90), indicating robust internal consistency. However, the barriers variable demonstrated a slightly lower reliability but was acceptable, with a Cronbach’s alpha value of 0.67, suggesting moderate reliability that warrants cautious interpretation in subsequent analyses.

Table 2 summarizes the descriptive statistics for the outcome variables (awareness, implementation, barriers, and total scale), including mean (M), standard deviation (SD), skewness, and kurtosis values.

In order to detect univariate outliers, the demographic data and self-report measures were transformed into standardized z-scores. The z-score is a statistical metric that quantifies the number of standard deviations by which a given value deviates from the mean of a given distribution. A total of six cases were discovered to have z-scores over 3.29, which is a typical cut-off for identifying outliers. However, to mitigate the impact, the outliers were constrained using the winsorization approach [28]. Winsorization refers to the statistical technique of restricting extreme values within a dataset to mitigate the potential impact of outliers that may be erroneous or anomalous. These psychometric properties suggest that the variables are reliable and normally distributed, making them suitable for further statistical analyses.

### 3.2. Sociodemographic Characteristics

The sociodemographic characteristics of 365 healthcare professionals in Saudi Arabia, a study population, revealed several notable demographic trends, as shown in Table 3. The gender distribution was predominantly male (55.5%, *n* = 203), with females comprising 44.5% (*n* = 163). The professional’s average age in this study was 31.31, with a standard deviation of 5.41 years.

Regarding educational qualifications and specialization, a significant majority hold a BSc degree, making up 68.6% (*n* = 251) of the sample, while the remaining 31.4% (*n* = 115) possess MSc or post-MSc degrees. Musculoskeletal Physical Therapy is the most prevalent specialty, representing 36.1% (*n* = 132) of the respondents, followed by Neurology Physical Therapy at 27.6% (*n* = 101). Experience-wise, 45.1% (*n* = 165) of the participants have 5–10 years of experience, compared to 37.2% (*n* = 136) with less than five years.

Regarding workplace settings, nearly half of the respondents, 49.5% (*n* = 181), work in hospitals, while 25.1% (*n* = 92) are in private practice. Geographically, the largest share of participants were from the Central Region of Saudi Arabia, accounting for 43.2% (*n* = 158).

The findings highlight a concentration of young, mid-career professionals in hospitals specializing in the musculoskeletal and neurology fields, which may impact the awareness and implementation of sports injury prevention programs in Saudi Arabia.

### 3.3. Awareness, Implementation, and Potential Barriers

Table 4 presents the responses of physical therapists regarding their awareness, implementation, and potential barriers related to sports injury prevention programs, categorized into five options, ranging from “Strongly Disagree” to “Strongly Agree”. Each item is analyzed in terms of frequency and percentage of responses, mean score, standard deviation, and the resulting decision based on the mean. A significant proportion of therapists strongly agreed that they are familiar with sports injury prevention programs (44.8%) and that such programs effectively reduce injuries among athletes (64.8%). However, when it comes to awareness of standard sports injury prevention programs, such as the FIFA 11+, KIPP, and the PEP Program, the mean was 3.53, and for implementing specific injury prevention programs, like the FIFA 11+, KIPP, and the PEP Program, the majority had neutral-to-low levels of agreement, reflecting a mean score of 3.12. The mean scores for items regarding implementation, such as regular incorporation of injury prevention strategies in treatment plans (M = 3.57), collaboration with other professionals (M = 3.33), and conducting pre-participation screenings (M = 3.08), were also on the lower side.

### 3.4. Comparison of the Outcome Scores of Awareness, Implementation, Barriers, and Total Scale Between Different Specialties of Physical Therapists

Table 5 compares the outcome scores for awareness, implementation, and barriers across different specialties of physical therapists in the Kingdom of Saudi Arabia (N = 366). The results are detailed below.

#### 3.4.1. Awareness: Significant Differences in Awareness Scores Were Observed Among the Various Specialties

Musculoskeletal Physical Therapy (N = 132) had a mean rank of 187.46, while Neurology Physical Therapy (N = 101) scored 133.37. Cardiopulmonary Physical Therapy (N = 14) had the lowest mean rank at 97.50, whereas Sports Physical Therapy (N = 85) achieved the highest score with a mean rank of 264.83. Pediatric Physical Therapy (N = 26) scored 161.38, and Women’s Health Physical Therapy (N = 8) had a mean rank of 109.25. These differences were found to be statistically significant.

#### 3.4.2. Implementation

Physical Therapy (N = 132) had a mean rank of 208.14, while Neurology Physical Therapy (N = 101) received a lower mean rank of 113.40. Cardiopulmonary Physical Therapy (N = 14) scored 107.54, and Sports Physical Therapy (N = 85) recorded the highest mean rank of 262.86. Pediatric Physical Therapy (N = 26) scored 142.65, whereas Women’s Health Physical Therapy (N = 8) received the lowest score of 84.44. These results highlight notable differences in implementation effectiveness across the specialties.

#### 3.4.3. Barriers

This study compared the different types of physical therapy based on their mean ranks. Musculoskeletal Physical Therapy (N = 132) had a mean rank of 181.78, while Neurology Physical Therapy (N = 101) had a mean rank of 155.09. Cardiopulmonary Physical Therapy (N = 14) scored 177.07. Sports Physical Therapy (N = 85) achieved the highest mean rank at 221.25, followed closely by Pediatric Physical Therapy (N = 26) with a mean rank of 202.60. Women’s Health Physical Therapy (N = 8) had the lowest mean rank at 118.63. The differences among these groups were statistically significant, with χ^2^(5) = 22.388 and *p* < 0.001.

#### 3.4.4. Scale Total

The data present the mean ranks of various physical therapy specialties based on their total scores. Musculoskeletal Physical Therapy (N = 132) had a mean rank of 199.25, while Neurology Physical Therapy (N = 101) had a mean rank of 115.91. Cardiopulmonary Physical Therapy (N = 14) scored 104.57, and Sports Physical Therapy (N = 85) achieved the highest mean rank of 272.36. Pediatric Physical Therapy (N = 26) had a mean rank of 148.02, whereas Women’s Health Physical Therapy (N = 8) had the lowest mean rank at 86.25. The differences among these specialties were statistically significant (χ^2^ = 121.698, df = 5, *p* < 0.001). Sports Physical Therapy consistently received the highest scores across all dimensions, while Women’s Health Physical Therapy generally received the lowest scores. These findings indicate significant variations in awareness, implementation, and barriers across the different specialties.

### 3.5. Comparison of Different Regions in Saudi Arabia

Significant regional differences were observed in awareness scores, with the North Region achieving the highest mean rank. However, no significant differences were found in implementation, barriers, or total scores across regions.

#### 3.5.1. Awareness

Significant differences were observed in awareness scores across the different regions. The Central Region (N = 158) had a mean rank of 176.31, while the East Region (N = 65) had a mean rank of 163.02. The West Region (N = 67) recorded a mean rank of 208.76; the North Region (N = 41) had the highest mean rank at 212.05. The South Region (N = 35) had a mean rank of 172.19.

#### 3.5.2. Implementation

There were no significant differences in implementation scores across the regions. The Central Region (N = 158) had a mean rank of 179.06, while the East Region (N = 65) had a mean rank of 160.05. The West Region (N = 67) scored a mean rank of 203.88, and the North Region (N = 41) achieved the highest mean rank of 205.98. The South Region (N = 35) had a mean rank of 181.74. The North Region had the highest implementation score, while the other regions showed comparable scores without significant differences.

#### 3.5.3. Barriers

No significant differences were found in barrier scores across regions. The Central Region (N = 158) had a mean rank of 193.95, the East Region (N = 65) had a mean rank of 172.98, and the West Region (N = 67) had a mean rank of 180.66. The North Region (N = 41) had a mean rank of 192.11, while the South Region (N = 35) had a mean rank of 173.23.

#### 3.5.4. Scale Total

No significant differences were found in the total scores across regions (χ^2^ = 9.090, df = 4, *p* = 0.059). The Central Region (N = 158) had a mean rank of 179.26, while the East Region (N = 65) had a mean rank of 160.53. The West Region (N = 67) recorded a mean rank of 203.60, and the North Region (N = 41) had the highest mean rank at 212.20. Additionally, the South Region (N = 35) had a mean rank of 173.23. Table 6 shows the details.

### 3.6. Comparison of the Level of Qualification

Comparison of the outcome scores of awareness, implementation, barriers, and total scale revealed significant differences (*p* < 0.05, 95% CI) between the level of qualifications (BSc and MSc or higher) of the physical therapists in Saudi Arabia.

#### 3.6.1. Awareness

The physical therapists with MSc or higher qualifications scored significantly (t = −2.26, *p* = 0.012) higher than those with a BSc, accompanied by a small effect size (Cohen’s d = 0.25).

#### 3.6.2. Implementation

Those with MSc or higher qualifications scored significantly (t = −2.15, *p* = 0.016) higher than BSc holders, with a small effect size (Cohen’s d = 0.24).

#### 3.6.3. Barriers

A higher score was observed among MSc or higher qualified physical therapists compared to those with a BSc (t = −2.26, *p* = 0.012, and the effect size was small (Cohen’s d = 0.25).

#### 3.6.4. Total Scale

The total scale score also showed a significantly higher difference for MSc or higher qualified individuals compared to BSc holders (t = −2.47, *p* = 0.007), with a small effect size (Cohen’s d = 0.28). These findings suggest that the physical therapist’s level of qualifications is associated with higher scores in awareness, implementation, barriers, and the scale total with small effect sizes, as shown in Table 7.

### 3.7. Correlation Between Study Significant Variables

A bivariate Pearson correlation was conducted to analyze the correlation between awareness, implementation, and barriers of SIPP among all specialties of physical therapists in Saudi Arabia.

#### 3.7.1. Awareness and Implementation

The two-tailed *p*-value is less than 0.01 (*p* = 0.000), suggesting that the correlation is statistically significant. This is a strong positive correlation (r = 0.723), indicating that implementation increases as the awareness of SIPP increases, as shown in Table 8.

#### 3.7.2. Awareness and Barriers

The two-tailed *p*-value is less than 0.01 (*p* = 0.000), suggesting that the correlation is statistically significant. This weak positive correlation (r = 0.270) suggests a slight increase in barriers as the awareness of SIPP increases, as shown in Table 8.

#### 3.7.3. Implementation and Barriers

The two-tailed *p*-value is less than 0.01 (*p* = 0.000), suggesting that the correlation is statistically significant. This represents a weak positive correlation (r = 0.280), suggesting that there is a slight increase in barriers as the implementation increases, as shown in Table 8.

## 4. Discussion

The findings of this study suggest that physical therapists in Saudi Arabia have a good awareness of sports injury prevention with statistically significant differences in awareness scores across various physical therapy specialties (Chi-Square χ2 = 87.934, df = 5, *p* < 0.001), highlighting variability in knowledge and understanding among practitioners in different domains. This discovery aligns with prior research documenting the awareness of injury prevention programs among French- and German-speaking sports medicine communities in Europe [27]. Evaluating the current state of sports injury prevention perception knowledge and practice revealed that reported knowledge was very high in most cases; 91% and only 54% reported being aware of specific injury prevention programs [28]. In similar research conducted on physical therapists worldwide, a total of 287 (59.3%) of the participants were aware of the current SIPPs, and 177 (36.6%) were implementing SIPPs. Their current findings of the study indicate that physiotherapists’ have average awareness and a low level of implementation of SIPPs [24,25], and more studies aim to assess the implementation of the OSTRC Knee Injury Prevention Program among professional basketball, handball, soccer, and volleyball players in the Gulf Cooperation Council (GCC). Countries concluded that implementing OSTRC knee injury prevention exercises was low among participants and was not statistically different between GCC countries or sports. These studies suggest that some common barriers and challenges hinder the implementation of sports injury prevention programs in different societies. It also suggests a notable difference in the degree of awareness and implementation of sports injury prevention programs among physical therapists specializing in various fields within Saudi Arabia. In particular, sports physical therapists have shown a greater degree of awareness and implementation in comparison to therapists specializing in other areas, such as musculoskeletal physical therapists, neurology physical therapists, cardiopulmonary physical therapists, pediatric physical therapists, and women’s health physical therapists. The above research findings imply that sports therapists are more aware of the need to implement sports injury prevention programs within their professional practice. This correlation may be due to their specialized training, educational background, or personal inclination toward their domain. The observation that sports physical therapists exhibit a greater degree of awareness and implementation of sports injury prevention programs compared to therapists in other disciplines aligns with prior research investigating the awareness of injury prevention programs in sports medicine communities across Western Europe and involved in injury prevention [29]. It was revealed that sports physical therapists reported better awareness of existing injury prevention programs and reported dedicating more time to injury prevention compared to surgeons [28].

Nevertheless, the findings suggest that sports physical therapists exhibited more significant obstacles than other disciplines. This discrepancy may indicate the unique problems and complexities sports physical therapists face when implementing sports injury prevention programs within their respective contexts. The reported implementation barriers included the need for standardized guidelines, limited resources, and a lack of athletic compliance and cultural factors. In contrast, injury prevention implementation challenges are widely known and discussed in the literature [30,31]. This feature is of utmost importance as it serves as a first measure to close the gap between the existing evidence, the awareness possessed by relevant stakeholders, and, subsequently, their attitudes toward the implementation process [5,32,33]. Nevertheless, it is somewhat unexpected and goes against the anticipated notion that sports physical therapists would encounter more favorable circumstances and receive greater support for implementing sports injury prevention programs, as indicated by the finding that they reported slightly higher barriers than therapists in other specialties. This discovery suggests that sports physical therapists encounter more difficulties and barriers within their work environments. A study on the barriers to injury prevention in professional football in Qatar disclosed that the execution of injury prevention measures has been revealed. The doctors affiliated with sports teams mostly perceive their role as being primarily engaged in treating and recovering athletes, with a lesser emphasis on involvement in preventative measures. The fundamental tasks of physiotherapists are the identification and assessment of injury risks, as well as the provision of personalized training regimens to athletes. The participants asserted that the responsibility for implementing injury prevention lies with the training coach. According to all stakeholders, the training coach is crucial in facilitating communication between the head coach and the medical staff [34]. Injury prevention is a collaborative effort involving multiple individuals working together. The success of the injury prevention implementation process is attributed to the staff members, who have significant responsibility. Each member of the team is assigned specific tasks and responsibilities. One participant noted that the failure to execute a task adequately disrupts the system’s integrity, resulting in a compromised efficacy of injury prevention measures [35].

Moreover, the findings indicated statistically significant regional differences in awareness scores, but for implementation, barriers, and overall scores, no statistically significant differences were observed across regions. Nevertheless, the findings revealed specific trends and patterns that imply potential regional variations in these variables among healthcare workers throughout Saudi Arabia. However, the findings of this study have uncovered certain inconsistencies with prior scholarly works that have examined the awareness and use of current sports injury prevention programs among physical therapists worldwide. The participants were characterized into six groups. Generally, there were statistically significant differences between the six regions in the awareness of 11+, 11+K, and KIPP.

On the other hand, there was not a statistically significant difference between the physiotherapy confederation regions in the awareness of 11+R, 11+S, GAA15, PEP, and Boksmart. According to the different physiotherapy confederation regions, there was a statistically significant difference only in the 11+S, the highest among the African physiotherapists at 14.8% and the lowest among North and Central America [25]. This difference could be due to the variety of samples and differences in culture and health education systems. On the other hand, Saudi culture, language, and education system are similar.

One of the goals of this study was to examine the level of awareness, implementation, and potential barriers to sports injury prevention programs among physical therapists with various levels of qualification in Saudi Arabia. The present study posited that physical therapists possessing higher qualifications, such as a Master of Science degree or higher, would exhibit greater levels of awareness and execution of sports injury prevention programs compared to their counterparts with lower qualifications, such as a Bachelor of Science degree. The findings of this research largely corroborated the proposed theory, as statistically significant disparities were observed between the two cohorts across all key study variables, namely awareness, implementation, barriers, and the scale total. The present study finding suggests a positive correlation between the qualification degree and better awareness, implementation, barriers, and scale total scores. The observed phenomenon can be attributed to the notion that physical therapists possessing higher degrees have a greater depth of knowledge and proficiency than their counterparts with lower qualifications. Physical therapists with advanced qualifications may also possess greater exposure and familiarity with the most current evidence. This can be attributed to their potential research, education, or professional development involvement. Furthermore, it is plausible that physical therapists possessing advanced degrees may exhibit increased self-assurance and drive when it comes to the implementation of sports injury prevention programs. This may be attributed to their heightened perception of the advantages of such programs. The present results align with another study’s aim to assess the association between educational status and awareness of adherence to preventive measures for COVID-19 in Saudi Arabia and revealed that higher education levels correlated with better awareness and adherence [36]. Higher-educated individuals often have a greater capacity for awareness and implementation due to their exposure to knowledge and critical thinking skills. Their education equips them with the tools to analyze complex issues, consider multiple perspectives, and make informed decisions. This heightened awareness extends beyond academic subjects and into various aspects of life.

Furthermore, a significant positive correlation existed between awareness and implementation (r = 0.723). This finding suggests that implementation tends to increase as the awareness increases. A study focused on the awareness and implementation of common sports injury prevention programs among physical therapists worldwide revealed that 59.3% of the participants were aware of the current SIPPs, and 177 (36.6%) were implementing SIPPs in their current practice, which was considered a low implementation level that is consistent with this study’s findings [23]. On the other hand, the correlation of awareness and implementation with barriers was evident with a weak positive correlation, even though it was a significant relationship in this study. However, it could also suggest that as the awareness increases, barriers are recognized, and the implementation of SIPPs is limited. Understanding this relationship is vital for stakeholders, highlighting the need for continuous support and adaptation to implement SIPPs.

### 4.1. Clinical Implications

To ensure the effective implementation of injury prevention programs, it is essential to address the identified barriers, which encompass the requirement for standardized guidelines or protocols and the issue of athlete compliance in the context of sports injury prevention in Saudi Arabia. This requires implementing measures in both the organizational and educational domains.This research’s findings possess the capacity to offer significant insights for educational institutions, professional organizations, and healthcare settings in Saudi Arabia. These findings have the potential to contribute to the advancement of targeted interventions, training initiatives, and policies designed to facilitate the prevention of sports-related injuries.Physical therapists, regardless of their level of proficiency or gender, may benefit from regular training updates and seminars specifically aimed at sports injury prevention. These opportunities enable them to stay up to date with the current knowledge and improve their skills.

### 4.2. Limitations

This study’s findings have limited applicability to physical therapists in Saudi Arabia as they do not specifically address sports-specialized physiotherapists. These findings may not be easily generalizable to other nations or cultures due to potential variances in educational systems, cultural views, and behaviors.Most participants in the study had a bachelor’s degree and specialized knowledge in orthopedics, incorporating years of experience. The limited range of variability among the participants may have constrained the findings and introduced a potential bias toward a specific knowledge base or skill set.The research mostly relies on self-reporting, a methodology vulnerable to subjective biases, and may not sufficiently capture real-world activities.

## 5. Conclusions

This research study reveals that physical therapists in Saudi Arabia are generally aware of sports injury prevention programs (SIPPs). However, their awareness and implementation of sports injury prevention programs (SIPPs) levels vary significantly across the specialties. Sports physical therapists have significantly higher awareness and implementation than other specialties; those with a master’s degree or higher show greater awareness than bachelor’s degree holders. There are minimal regional differences, and a strong positive correlation exists between awareness and implementation. For future perspective, it is essential to provide targeted training programs, increase access to educational resources, and promote institutional support to enhance the implementation of SIPPs in rehabilitation settings. Public awareness campaigns and cross-specialty collaborations should be encouraged to play a vital role in bridging the gap between awareness and practical application, ultimately improving injury prevention practices in Saudi Arabia.

## Figures and Tables

**Table 1 medicina-61-00121-t001:** Details of content validity.

Domain Item	Ex1	Ex2	Ex3	Ex4	Ex5	Ex6	Expert in Agreement	I-CVI Item Content Validity Index	UI Universal Agreement
1	4	4	4	4	4	4	6	1	1
2	4	4	4	4	4	4	6	1	1
3	4	4	4	4	4	4	6	1	1
4	4	4	4	4	4	3	6	1	1
5	4	4	4	4	4	4	6	1	1
6	4	4	1	4	4	4	5	0.83	0
7	4	4	4	4	4	4	6	1	1
8	4	4	4	4	4	4	6	1	1
9	4	4	4	4	4	4	6	1	1
10	4	4	4	4	4	4	6	1	1
11	4	4	4	4	4	4	6	1	1
12	4	4	4	4	4	4	6	1	1
13	4	4	4	3	4	4	6	1	1
14	4	4	4	2	4	4	5	0.83	0
15	4	4	4	4	4	4	6	1	1
16	4	4	4	4	4	4	6	1	1
17	4	4	4	4	4	4	6	1	1
18	4	4	4	4	4	4	6	1	1
19	4	4	4	4	4	4	6	1	1
20	4	4	4	4	4	4	6	1	1
21	4	4	4	3	4	4	6	1	1
22	4	4	4	4	3	4	6	1	1
23	4	4	4	4	4	4	6	1	1
24	4	4	4	4	4	4	6	1	1
25	4	4	4	4	4	4	6	1	1
S-CVI Average (S-CVI Ave)								0.98	S-CVI Universal Agreement (S-CVI UA Ave) = 0.92

S-CVI Average (S-CVI Ave): 0.98; S-CVI Universal Agreement (S-CVI UA) Ave: 0.92; Ex = expert; I-CVI = item content validity index; UI = universal agreement; S-CVI = scale content validity index.

**Table 2 medicina-61-00121-t002:** Details of the psychometric characteristics of the variables.

Range								
Variable	k	α	m	SD	Potential	Actual	Skew	Kurt
Awareness	7	0.83	28.53	4.49	7–35	16–35	−0.468	−0.225
Implementation	7	0.90	23.83	6.17	7–35	7–35	−0.197	−0.511
Barriers	4	0.67	15.56	2.54	4–20	10–20	−0.234	−0.413
ScaleTotal	18	0.90	67.80	11.25	18–90	32–90	−0.332	0.175

Note k = key variable numbers; α = Cronbach’s alpha; SD = standard deviation.

**Table 3 medicina-61-00121-t003:** Details of sociodemographic characteristics.

Variable	Category	f	%
Gender	Male	203	55.5
	Female	163	44.5
Place of Work	Hospital	181	49.5
	Rehabilitation Center	59	16.1
	Sports Clinic	34	9.3
	Private Practice	92	25.1
Years of Experience	Less than 5 Years	136	37.2
	5–10 Years	165	45.1
	10 Years and Above	65	17.8
Qualification	BSc	251	68.6
	MSc and Post-MSc	115	31.4
Specialty	Musculoskeletal Physical Therapy	132	36.1
	Neurology Physical Therapy	101	27.6
	Cardiopulmonary Physical Therapy	14	3.8
	Sports Physical Therapy	85	23.2
	Pediatric Physical Therapy	26	7.1
	Women’s Health Physical Therapy	8	2.2
Region in SA	Central Region	158	43.2
	East Region	65	17.8
	West Region	67	18.3
	North Region	41	11.2
	South Region	35	9.6

**Table 4 medicina-61-00121-t004:** Responses of physical therapists on awareness, implementation, and barriers of sports injury prevention programs.

S.N	Items	Sd *n* (%)	D *n* (%)	N *n* (%)	A *n* (%)	Sa *n* (%)	m	SD	Decision
1	I am familiar with the concept of sports injury prevention programs.	6 (1.6)	4 (1.1)	49 (13.4)	143 (39.1)	164 (44.8)	4.24	0.85	
2	I have received specific training or education on sports injury prevention programs.	17 (4.6)	46 (12.6)	84 (23.0)	115 (31.4)	104 (28.4)	3.66	1.15	Low
3	I am aware of the different components involved in sports injury prevention programs (e.g., warm-up exercises, strength training, flexibility training, injury management).	5 (1.4)	7 (1.9)	34 (9.3)	147 (40.2)	173 (47.3)	4.30	0.82	
4	I believe that sports injury prevention programs are effective in reducing the risk of injuries among athletes.	5 (1.4)	3 (0.8)	18 (4.9)	103 (28.1)	237 (64.8)	4.54	0.75	
5	I regularly update my knowledge and skills in sports injury prevention through professional development activities (e.g., workshops, conferences).	12 (3.3)	43 (11.7)	87 (23.8)	97 (26.5)	127 (34.7)	3.78	1.14	
6	I am aware of common sports injury prevention programs, such as the FIFA 11+, KIPP, and the PEP Program.	23 (6.3)	50 (13.7)	96 (26.2)	105 (28.7)	92 (25.1)	3.53	1.19	Low
7	I believe that increasing awareness and education about sports injury prevention among athletes, coaches, and parents is essential.	2 (0.5)	6 (1.6)	31 (8.5)	133 (36.3)	194 (53.0)	4.40	0.76	
8	I regularly implement sports injury prevention programs, such as the FIFA 11+, KIPP, and the PEP Program.	31 (8.5)	83 (22.7)	114 (31.1)	87 (23.8)	51 (13.9)	3.12	1.16	Low
9	I am confident in my ability to assess and identify biomechanical factors that contribute to sports injuries.	13 (3.6)	46 (12.6)	85 (23.2)	137 (37.4)	85 (23.2)	3.64	1.08	Low
10	I regularly incorporate sports injury prevention strategies into my treatment plans for athletes.	18 (4.9)	44 (12.0)	97 (26.5)	126 (34.4)	81 (22.1)	3.57	1.11	Low
11	I have access to appropriate resources and guidelines for implementing sports injury prevention programs.	12 (3.3)	47 (12.8)	105 (28.7)	127 (34.7)	75 (20.5)	3.56	1.05	Low
12	I collaborate with other healthcare professionals (e.g., sports physicians, coaches) to develop and implement sports injury prevention programs.	26 (7.1)	53 (14.5)	124 (33.9)	101 (27.5)	62 (16.9)	3.33	1.13	Low
13	I provide educational materials and advice to athletes on injury prevention techniques.	20 (5.5)	32 (8.7)	112 (30.6)	140 (38.3)	62 (16.9)	3.52	1.05	Low
14	I conduct pre-participation screenings or assessments to identify athletes at risk of injury.	41 (11.2)	79 (21.6)	104 (28.4)	92 (25.1)	50 (13.7)	3.08	1.21	Low
15	I believe that cultural factors in Saudi Arabia influence the implementation of sports injury prevention programs.	13 (3.6)	19 (5.2)	85 (23.2)	154 (42.1)	95 (26.0)	3.82	0.99	
16	There is a need for standardized guidelines or protocols for sports injury prevention in Saudi Arabia.	5 (1.4)	10 (2.7)	71 (19.4)	122 (33.3)	158 (43.2)	4.14	0.92	
17	Limited resources (e.g., equipment, facilities) hinder the implementation of sports injury prevention programs in my practice.	4 (1.1)	39 (10.7)	108 (29.5)	136 (37.2)	79 (21.6)	3.67	0.97	Low
18	Lack of athlete compliance or motivation poses challenges in implementing sports injury prevention programs.	2 (0.5)	20 (5.5)	87 (23.8)	165 (45.1)	92 (25.1)	3.89	0.86	

Note: Sd = Strongly disagree; D = Disagree; N = Neutral; A = Agree; Sa = Strongly agree; Decision-Weighted Average 67.79/18 = 3.77.

**Table 5 medicina-61-00121-t005:** Comparison of the outcome scores of awareness, implementation, and barriers between different specialties of physical therapists in Saudi Arabia using a Kruskal–Wallis test (N = 366).

Variable	Group	N	Mean Rank	Chi-Square	df	*p*
Awareness	Musculoskeletal Physical Therapy	132	187.46	87.934	5	<0.001
	Neurology Physical Therapy	101	133.37			
	Cardiopulmonary Physical Therapy	14	97.50			
	Sports Physical Therapy	85	264.83			
	Pediatric Physical Therapy	26	161.38			
	Women’s Health Physical Therapy	8	109.25			
Implement	Musculoskeletal Physical Therapy	132	208.14	117.792	5	<0.001
	Neurology Physical Therapy	101	113.40			
	Cardiopulmonary Physical Therapy	14	107.54			
	Sports Physical Therapy	85	262.86			
	Pediatric Physical Therapy	26	142.65			
	Women’s Health Physical Therapy	8	84.44			
Barriers	Musculoskeletal Physical Therapy	132	181.78	22.388	5	<0.001
	Neurology Physical Therapy	101	155.09			
	Cardiopulmonary Physical Therapy	14	177.07			
	Sports Physical Therapy	85	221.25			
	Pediatric Physical Therapy	26	202.60			
	Women’s Health Physical Therapy	8	118.63			
Total Scale	Musculoskeletal Physical Therapy	132	199.25	121.698	5	<0.001
	Neurology Physical Therapy	101	115.91			
	Cardiopulmonary Physical Therapy	14	104.57			
	Sports Physical Therapy	85	272.36			
	Pediatric Physical Therapy	26	148.02			
	Women’s Health Physical Therapy	8	86.25			

**Table 6 medicina-61-00121-t006:** Comparison of the outcome scores of awareness, implementation, and barriers between different regions in Saudi Arabia (N = 366).

Variable	Group	N	Mean Rank	Chi-Square	df	*p*
Awareness	Central Region	158	176.31	10.433	4	0.034 *
	East Region	65	163.02			
	West Region	67	208.76			
	North Region	41	212.05			
	South Region	35	172.19			
Implement	Central Region	158	179.06	7.842	4	0.098
	East Region	65	160.05			
	West Region	67	203.88			
	North Region	41	205.98			
	South Region	35	181.74			
Barriers	Central Region	158	193.95	5.853	4	0.210
	East Region	65	172.98			
	West Region	67	180.66			
	North Region	41	192.11			
Scale Total	Central Region	35	179.26	9.090	4	0.059
	East Region	158	160.53			
	West Region	65	203.60			
	North Region	67	212.20			
	South Region	41	173.23			

* it indicate the Significant at *p*-value < 0.05.

**Table 7 medicina-61-00121-t007:** Comparison of the outcome scores of awareness, implementation, and barriers between the levels of qualification of physical therapists in Saudi Arabia (N = 366).

Variables	BSc (*n* = 251)		MSc and Post-MSc (*n* = 115)		95% Confidence Interval	
	Mean Rank	Sum of Ranks	Mean Rank	Sum of Ranks	Mann–Whitney U	*p*
Awareness	172.40	43,273.00	207.72	23,888.00	11,647.000	0.003 *
Implementation	175.53	44,058.00	200.90	23,103.00	12,432.000	0.033 *
Barriers	175.45	44,038.50	201.07	23,122.50	12,412.500	0.030 *
Scale Total	173.27	43,490.00	205.83	23,671.00	11,864.000	0.006 *

* it indicate the significant at *p*-value < 0.05.

**Table 8 medicina-61-00121-t008:** Correlation between awareness, implementation, and barriers of SIPP among all specialties of physical therapists in Saudi Arabia using Pearson’s correlation coefficient test (N = 366, 95% CI).

Variables		Awareness	Implementation	Barrier
Awareness	r	1	0.723 **	0.270 **
	p		0.000	0.000
Implementation	r	0.723 **	1	0.280 **
	p	0.000		0.000
Barrier	r	0.270 **	0.280 **	1
	p	0.000	0.000	

** Correlation is significant at the 0.01 level (two-tailed).

## Data Availability

The original contributions presented in this study are included in the article. Further inquiries can be directed to the corresponding authors.

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
