# Peer review of "Awareness and Implementation of Sports Injury Prevention Programs Among Physical Therapists in Saudi Arabia: A Cross-Sectional Study"

_medicina, 2025, doi:10.3390/medicina61010121_

Round 1

Reviewer 1 Report

Comments and Suggestions for Authors

This study is very interesting and addressing an important gap that is currently existing in literature. Assessing and analysing injury prevention practices as well as barriers to implementation is crucial if we want to understand the situation in a specific context. 

The manuscript is well written but has some limits :

- the questionnaire (and the entire manuscript) only focus on 3 prevention programs, while other have been proven as efficient

- years of experience are not considered in the analysis

- the rationale of including physiotherapists that are not specialized in sports physio (and are probably not working with sportspeople) is not clear ...

- The discussion should describe some clinical and future perspectives. The results are summarized but we don't see what are the priorities to face this situation and improve injury prevention in the future.

Some other comments are written in the text.

Author Response

Thank you very much for taking the time to review this manuscript. Please find the detailed responses below and the corresponding revisions/corrections highlighted in the re-submitted files.

Comments and Suggestions for Authors: This study is very interesting and addressing an important gap that is currently existing in literature. Assessing and analysing injury prevention practices as well as barriers to implementation is crucial if we want to understand the situation in a specific context. 

Response: Thank you for your insightful comments and suggestions. We are glad you found our study interesting and acknowledged its relevance in addressing a critical gap in the literature. We agree that assessing and analyzing injury prevention practices and barriers to implementation is essential for understanding the situation within specific contexts.

Comments - The manuscript is well written but has some limits : the questionnaire (and the entire manuscript) only focus on 3 prevention programs, while other have been proven as efficient

Response: Thank you for your feedback. We recognize that other programs have demonstrated effectiveness. However, our focus on FIFA 11+, KIPP, and PEP was based on their widespread implementation and prominence in the existing literature. Future studies will include additional programs, such as Funball and OSTRC, for a more comprehensive analysis. We have noted this as a limitation in the discussion section.Line:451-460

Comments :- years of experience are not considered in the analysis

Response: Thank you for insight. Years of experience are essential for evaluating familiarity with SIPPs. This data was already included in in Table 3, representing the sociodemographic characteristics. Although, we didn’t analyse it separately as a comparing variable, so mentioned it in the limitation of the study (Line: 508-509).

Comments :- the rationale of including physiotherapists that are not specialized in sports physio (and are probably not working with sportspeople) is not clear ...

Response: The inclusion of non-sports physiotherapists aimed to offer broader insights into awareness, implementation, and barrier to follow SIPPs while managing the athletes. It has been added in the revised manuscript (Line: 119-122).

Comments: - The discussion should describe some clinical and future perspectives. The results are summarized but we don't see what are the priorities to face this situation and improve injury prevention in the future.

Response: Thank you for your valuable suggestion. We have incorporated some specific clinical Implications (Line: 497-510) and future perspectives (Line: 529-535).

Comments: Some other comments are written in the text.

Response: The pdf version has been reviewed and Answered.

Reviewer 2 Report

Comments and Suggestions for Authors

The aim of this study was to evaluate physiotherapists' awareness of and implementation of Sports Injury Prevention Programs (SIPPs) in Saudi Arabia.

The current study has several needs to be concerned. For example;

Abstract and Introduction Issues:

1.        The number of words in the abstract exceeds the word limit as listed in the journal guidelines.

2.        The introduction is lengthy, and concepts like the benefits of physical activity and the risks of sports injuries are reiterated multiple times without providing substantial new information. The introduction would be stronger if these points were presented more concisely and without unnecessary repetition. Please consider rearranging the introduction to make it more concise and focused.

3.        While the motivation to study Saudi physiotherapists is mentioned, the text lacks specific data or preliminary observations from Saudi Arabia that could better justify the research need. Including regional statistics, preliminary studies, or anecdotal evidence from Saudi Arabia would strengthen the rationale for the study.

4.        The claims about the awareness and implementation of SIPPs globally are too broad and lack detailed analysis, particularly for regions outside Europe and Asia. No details of the items of SIPP are introduced in the introduction.

5.        Some references are missing. The references should be more cohesively woven into the narrative to build a stronger, more connected argument. References should be linked directly to specific claims rather than being listed at the end of the full stop.

Methodological Issues:

6.        The manuscript provides a sample size calculation but fails to explain each parameter (e.g., N, e, p, z). A clearer description of these parameters is needed.

7.        The section on ethical considerations (2.6) should be merged into Section 2.1 (Research Design) to improve the clarity and flow of the manuscript. Having separate sections for related concepts creates unnecessary fragmentation.

8.        The study uses ANOVA to analyze ordinal data (e.g., "agree," "neutral," "disagree"). Ordinal data have a natural order, but the intervals between categories are not equal, making it inappropriate to apply parametric tests like ANOVA. The use of a non-parametric test would be more suitable for analyzing ordinal data. The inappropriate statistical analysis is a critical flaw in the study's methodology.

Author Response

Point-by-point response to the reviewer1’s comments: Round 1

[Manuscript ID: medicina-3361880]

Reviewer 2:

Thank you very much for taking the time to review this manuscript. Please find the detailed responses below and the corresponding revisions/corrections highlighted in the re-submitted files.

Comments and Suggestions for Authors

The aim of this study was to evaluate physiotherapists' awareness of and implementation of Sports Injury Prevention Programs (SIPPs) in Saudi Arabia.

The current study has several needs to be concerned. For example;

Abstract and Introduction Issues:

  1. Comments: The number of words in the abstract exceeds the word limit as listed in the journal guidelines.

Response: Thank you for pointing this out. We have revised the abstract to comply with the journal's word limit (300 words) while ensuring that all key findings and conclusions are retained.

  1. Comments: The introduction is lengthy, and concepts like the benefits of physical activity and the risks of sports injuries are reiterated multiple times without providing substantial new information. The introduction would be stronger if these points were presented more concisely and without unnecessary repetition. Please consider rearranging the introduction to make it more concise and focused.

Response: We appreciate this feedback. The introduction has been revised to remove repetitive statements. It is now more concise, focused on the study's rationale, and directly aligned with the research objectives (Lines: 40-43; 55-63; & 83-86).

  1. Comments: While the motivation to study Saudi physiotherapists is mentioned, the text lacks specific data or preliminary observations from Saudi Arabia that could better justify the research need. Including regional statistics, preliminary studies, or anecdotal evidence from Saudi Arabia would strengthen the rationale for the study.

Response: Thank you for this valuable suggestion. We have incorporated the references (ref. 18, 22-23, 24, 25, 27) into preliminary studies on sports injuries and physiotherapy practices in Saudi Arabia. These additions strengthen the justification for conducting this study within the context of Saudi Arabia.

  1. Comments: The claims about the awareness and implementation of SIPPs globally are too broad and lack detailed analysis, particularly for regions outside Europe and Asia. No details of the items of SIPP are introduced in the introduction.

Response: Thank you for your feedback. We appreciate your constructive comments. We have added lines regarding the awareness and implementation of SIPPs from a global perspective and clarity in the introduction (Line:83-86; ref. 22-23).

  1. Comments: Some references are missing. The references should be more cohesively woven into the narrative to build a stronger, more connected argument. References should be linked directly to specific claims rather than being listed at the end of the full stop.

Response: We have thoroughly reviewed the manuscript and ensured appropriate references support all claims. References  (ref. 18, 22-23, 24, 25, 27) have been integrated more cohesively into the narrative to create a stronger and more logical flow of arguments.

Methodological Issues:

  1. Comments: The manuscript provides a sample size calculation but fails to explain each parameter (e.g., N, e, p, z). A clearer description of these parameters is needed.

Response: Thank you for highlighting this issue. We have clarified the sample size calculation parameters in the methods section, providing detailed explanations of each component (e.g., N = population size, e = margin of error, p = estimated proportion, and z = critical value for confidence level. [Line:110-113]

  1. Comments: The section on ethical considerations (2.6) should be merged into Section 2.1 (Research Design) to improve the clarity and flow of the manuscript. Having separate sections for related concepts creates unnecessary fragmentation.

Response: Thank you for your recommendation. The ethical considerations section has been merged into Section 2.1 (Research Design) to improve the clarity and flow of the methods section.

  1. Comments :The study uses ANOVA to analyze ordinal data (e.g., "agree," "neutral," "disagree"). Ordinal data have a natural order, but the intervals between categories are not equal, making it inappropriate to apply parametric tests like ANOVA. The use of a non-parametric test would be more suitable for analyzing ordinal data. The inappropriate statistical analysis is a critical flaw in the study's methodology.

Response: Thank you for highlighting your concerns about using ANOVA to analyze ordinal data. We understand the importance of choosing the proper statistical methods based on the data type. In response, we have reanalyzed the data using non-parametric methods, specifically the Kruskal-Wallis Test and the Mann-Whitney U Test, to ensure the validity of our findings. We have also revised the results and discussion sections of the manuscript accordingly.

 !!!Thank you again!!!

Round 2

Reviewer 2 Report

Comments and Suggestions for Authors

Thank you for submitting the revised version of your manuscript. The revisions appear to have mostly addressed the comments provided during the initial review.